# Impaired functions of human monocyte-derived dendritic cells and induction of regulatory T cells by pathogenic *Leptospira*

**Pratomporn Krangvichian[1], Teerasit Techawiwattanaboon[2,3], Tanapat Palaga[3,4], Patcharee Ritprajak[5], Patipark Kueanjinda[2], Chamraj Kaewraemruaen[6], Kanitha Patarakul[2,3]***

**1** Medical Microbiology, Interdisciplinary Program, Graduate School, Chulalongkorn University, Bangkok, Thailand, **2** Department of Microbiology, Faculty of Medicine, Chulalongkorn University and King Chulalongkorn Memorial Hospital, Bangkok, Thailand, **3** Chula Vaccine Research Center (Chula VRC), Center of Excellence in Vaccine Research and Development, Chulalongkorn University, Bangkok, Thailand, **4** Department of Microbiology, Faculty of Science, Chulalongkorn University, Bangkok, Thailand, **5** Research Unit in Integrative Immuno-Microbial Biochemistry and Bioresponsive Nanomaterials, Department of Microbiology, Faculty of Dentistry, Chulalongkorn University, Bangkok, Thailand, **6** Department of Science and Bioinnovation, Faculty of Liberal Arts and Science, Kasetsart University, Kamphaeng Saen Campus, Nakhon Pathom, Thailand

* kanitha.pa@chula.ac.th

**Data Availability Statement:** The data that support the finding of this study are publicly available from NCBI's Sequence Read Archive under accession

## Abstract

Leptospirosis is a global zoonosis caused by pathogenic *Leptospira*. The disease outcome is influenced by the interplay between innate and adaptive immune responses. Dendritic cells (DCs) play a crucial role in shaping the adaptive immune response. A recent study revealed that pathogenic *Leptospira* limited the activation of human monocyte-derived dendritic cells (MoDCs) compared to non-pathogenic *Leptospira*, but their impact on T-cell responses has not been investigated. Our study is the first to explore how viable pathogenic and non-pathogenic *Leptospira* affect the interaction between human MoDCs and T cells. We found that MoDCs infected with pathogenic leptospires (*L. interrogans* serovar Pomona and a clinical isolate, MoDCs-P) exhibited lower levels of CD80 and CD83 expression, suggesting partially impaired MoDC maturation, induced regulatory T cells (Tregs) while failing to induce CD4$^+$ T cell proliferation, compared to MoDCs infected with non-pathogenic leptospires (*L. biflexa* serovar Patoc and *L. meyeri* serovar Ranarum, MoDCs-NP). In contrast, non-pathogenic leptospires enhanced MoDC maturation and induced higher T cell proliferation including IFN-γ-producing CD4$^+$ T cells, indicative of a Th1-type response. Furthermore, pathogenic leptospires induced higher MoDC apoptosis through a cysteine aspartic acid-specific protease-3 (caspase-3)-dependent pathway and upregulated expression of the prostaglandin-endoperoxide synthase 2 (*PTGS2*) gene. Notably, prostaglandin E2 (PGE2), a product of the PTGS2 pathway, was found at higher levels in the sera of patients with acute leptospirosis and in the supernatant of MoDCs-P, possibly contributing to Treg induction, compared to those of healthy donors and MoDCs-NP, respectively. In conclusion, this study reveals a novel immunosuppressive strategy employed by pathogenic *Leptospira* to evade host immunity by partially impairing MoDC maturation and inducing Tregs. These

number PRJNA945908. You can access them through this URL: https://www.ncbi.nlm.nih.gov/sra/?term=PRJNA945908.

**Funding:** This work was supported by the Royal Golden Jubilee Ph.D. Scholarship of the Thailand Research Fund (grant no. PHD/0140/2559 to KP and PK); Chulalongkorn University (grant "the 90th Anniversary of Chulalongkorn University Fund" to KP and PK); the Ratchadapiseksompotch Fund, Faculty of Medicine, Chulalongkorn University (grant no. RA65/015 to KP). The funders had no role in the study design, data collection and analysis, decision to publish, or preparation of the manuscript.

**Competing interests:** The authors have declared that no competing interests exist.

findings deepen our understanding of leptospirosis pathogenesis in humans and may provide a novel strategy to modulate DCs for the prevention and treatment of the disease.

## Author summary

Dendritic cells (DCs) are critical antigen-presenting cells that link between innate and adaptive immune responses against pathogens. However, their role in the pathogenesis of human leptospirosis remains unclear. This study provides the first evidence of the impact of leptospires on MoDC-CD4$^+$ T cell responses. We observed that pathogenic leptospires partially impaired MoDC maturation, potentially compromising their ability to induce CD4$^+$ T cell proliferation while inducing regulatory T cells (Tregs). Notably, pathogenic leptospires upregulated the *PTGS2* gene expression, resulting in elevated prostaglandin E2 (PGE2) levels, which may contribute to Treg activation. Furthermore, pathogenic leptospires trigger more MoDC apoptosis, partly through a caspase-3-dependent pathway, compared to non-pathogenic strains. These findings might reveal the mechanisms that pathogenic leptospires used to evade the host immune response, facilitating their dissemination within the host. Conversely, non-pathogenic leptospires promote MoDC maturation and stimulate the proliferation of IFN-γ-producing CD4$^+$ T cells, potentially eliciting a Th1-type response. In conclusion, our study reveals a novel strategy employed by pathogenic leptospires for immune evasion while highlighting the immune-activating effects of non-pathogenic strains.

## Introduction

Leptospirosis is a re-emerging zoonosis and one of the major public health concerns. Leptospiral infection has a higher incidence in tropical and subtropical regions, and the estimated global incidence of leptospirosis is over 1 million cases and 60,000 deaths annually [1–3]. *Leptospira* spp. is recently classified into 4 subclades based on phylogenomic revolution and pathogenicity level [4,5]. Pathogenic strains (subclade P1) have commonly been reported to cause leptospirosis in humans. Intermediate strains (subclade P2) are unusual causes of leptospirosis and are more closely related to pathogenic strains. Non-pathogenic strains (subclades S1 and S2) are saprophytic species that naturally survive in the environment and are not known to cause leptospirosis. Pathogenic species colonize renal tubules of reservoir hosts and are excreted via urine, leading to contamination in environments. The pathogen enters the human body through skin abrasion and mucous membrane, leading to various clinical manifestations from mild to severe and life-threatening infections, such as kidney and hepatic failure and pulmonary hemorrhagic syndrome [6]. Pathogenic leptospires have previously been shown to evade host immune defenses, such as complement systems and innate immune recognition receptors [7–9], which is an important mechanism of their pathogenicity.

Vaccination with inactivated *Leptospira* primarily induces a humoral immune response against lipopolysaccharide (LPS) and provides protection [10]. In terms of adaptive immune protection, *L. borgpetersenii* killed vaccine induced a Th1 response and activated both CD4$^+$αβ and WC1$^+$γδ T proliferation [11]. It is likely that not only humoral-mediated immunity, but also cell-mediated immunity, are important immune responses against *Leptospira* infection. Dendritic cells (DCs) are professional antigen-presenting cells that play a central role in linking between innate and adaptive immune responses. The function of DCs is mainly to capture,

process, and present antigens to adaptive immune cells, thereby inducing their differentiation into effector cells. DCs go through two stages: immature and mature DC. Immature DCs are highly effective in capturing and processing antigens. DCs migrate from peripheral tissue to draining lymph nodes, where they can induce DC maturation and directly stimulate T cells. The types of T cell-mediated immune response depend not only on DC maturation, but also on activation signals received from the surrounding microenvironment, including cytokines [12]. Previous research has shown that both live virulent and attenuated pathogenic leptospires can bind to DC-SIGN, leading to maturation of monocyte-derived dendritic cells (MoDCs) and the production of IL-10, IL- 12p70, and TNF-α [13]. Moreover, a pathogenic strain of leptospires demonstrated reduced activation of human MoDCs compared to non-pathogenic leptospires, as indicated by the lower percentage of CD83+ cells. Notably, pathogenic leptospires induced a lower production of both pro- and anti-inflammatory cytokines than the non-pathogenic strain [14]. However, it is worth noting that these results were not consistent for all the pathogenic strains examined in that study. Furthermore, pathogenic and non-pathogenic leptospires activated DCs within 24 h after infection in C3H/HeJ mice [15]. Despite these findings, the impact of DCs infected with leptospires on T cells remains unclear. Therefore, this study aimed to compare the effects of viable pathogenic (*L. interrogans* serovar Pomona and a clinical isolate) and non-pathogenic (*L. biflexa* serovar Patoc and *L. meyeri* serovar Ranarum) leptospires on human MoDCs and their influence on CD4+ T cell responses. Our results reveal a novel strategy that may be utilized by pathogenic leptospires to evade host immune responses.

## Methods

### Ethical statement

The collection of buffy coats and serum samples was carried out according to institutional guidelines and approved by the Institutional Ethics Committee with the approval numbers: 12/2563, 189/63, and 675/63.

### Human buffy coats and serum samples

The buffy coats and sera of five healthy blood donors were obtained from the Thai Red Cross Society, Bangkok, Thailand. Five serum samples from patients with acute leptospirosis were collected at Hat Yai Hospital, Songkhla, Thailand.

### *Leptospira* and growth condition

In this study, low passage (< 5 *in vitro* passages) pathogenic strains (*Leptospira interrogans* serovar Pomona (PO) and a clinical isolate of *L. interrrogans* (CI), and two non-pathogenic strains, *L. biflexa* serovar Patoc strain Patoc I (PA) and *L. meyeri* serovar Ranarum (RA), were used. The clinical strain was isolated from the blood of a patient with leptospirosis at Surin Hospital, Surin, Thailand. Leptospires were cultured in liquid Ellinghausen-McCullough-Johnson-Harris (EMJH) medium (BD Difco) supplemented with 10% bovine serum albumin solution at 30˚C under aerobic conditions without agitation until the exponential growth phase was reached ($\sim 2{\times}10^8$ cells/ml) [16].

### Isolation of CD14+ monocytes and autologous naïve CD4+ T cells

Peripheral blood mononuclear cells (PBMCs) were isolated from buffy coats using the density gradient centrifugation method using lymphocyte separation medium (Biowest). CD14+ monocytes were separated from PBMCs by positive selection using CD14 antibody-coated

magnetic beads according to the manufacturer's instruction (Miltenyi Biotec). Autologous naïve CD4$^+$ T cells were separated from unlabeled CD14 monocytes by negative selection using the Mojosort Human CD4$^+$ T cell isolation kit (BioLegend) following the manufacturer's protocol. Naïve CD4$^+$ T cells were cryopreserved with RPMI 1640 containing 10% fetal bovine serum (FBS) (Gibco) before use.

## Generation and infection of human monocyte-derived dendritic cells

To generate immature MoDCs, CD14$^+$ monocytes ($1\times10^6$ cells/ml) were cultured in a culture medium (RPMI 1640 with 10% heat-inactivated FBS, 1% non-essential amino acids, 2 mM L-glutamine, 1 mM sodium pyruvate and 1% penicillin and streptomycin). Recombinant human granulocyte macrophage colony-stimulating factor (GM-CSF) and recombinant human interleukin 4 (IL-4) (PeproTech) were added to the culture at a final concentration of 25 ng/ml, and half of the culture medium was replaced every 2 days with a fresh culture medium containing GM-CSF and IL-4 [17]. Plates of immature MoDCs were incubated at 37˚C for 7 days in a humidified atmosphere containing 5% $CO_2$. This study used CD14$^+$ cells, MoDCs, and autologous naïve CD4$^+$ T cells with a purity of at least 90% (S1 Fig).

On day 7, MoDCs were detached from flasks by gently pipetting using cold phosphate buffered saline (PBS) and tapping the flasks firmly. Subsequently, the immature MoDCs were centrifuged at 500 ×g for 10 min and resuspended in the RPMI 1640 medium without antibiotics. Prior to seeding in the plates, CD1a and CD83 markers were assessed to ensure that cells remained as immature MoDCs. Then, MoDCs (at a density of $10^6$ cells/ml) were seeded in 24-well plates (Corning). Following that, the immature MoDCs were infected with leptospires at multiplicities of infection (MOIs) of 10, 50, and 100 in RPMI 1640 culture medium without antibiotics. Next, the plates were centrifuged at 500 ×g for 10 min to synchronize infection [18]. Immature MoDCs treated with 5 µg/ml of ultrapure lipopolysaccharide (LPS) from *Escherichia coli* K12 (Invitrogen) were used as a positive control and uninfected MoDCs were used as a negative control. After 24-h infection, the culture supernatant was collected and immediately stored at -80˚C to be used for measuring cytokine production. MoDCs were collected to determine cell viability, cell surface markers of maturation, and CD4$^+$ T cell differentiation.

## Evaluation of MoDC viability and apoptosis after infection with *Leptospira*

After 24-h infection, MoDC viability at an MOI of 100 was measured by the dimethylathiazol carboxymethoxyphenyl sulfophenyl tetrazolium (MTS) assay using the CellTiter 96 Aqueous Non-Radiative Cell Proliferation assay (Promega) according to the manufacturer's instruction. The absorbance intensity measured by a microplate reader (Thermo Fisher Scientific) at 490 nm. The percentage of cell viability was calculated as follows: [(Mean optical density (OD) of the sample–Mean OD of the medium) / (Mean OD of the control–Mean OD of the medium)] × 100%.

Cell apoptosis was evaluated using fluorescein isothiocyanate (FITC) Annexin V/7- aminoactinomycin D (AAD) staining according to the manufacturer's instruction (BioLegend) and assessed by flow cytometry. In brief, $1\times10^5$ immature MoDC cells were seeded into 48-well plates (Corning) and infected with pathogenic and non-pathogenic leptospires at MOIs of 10, 50, and 100 for 24 and 48 h. Immature MoDCs treated with 100 µM hydrogen peroxide ($H_2O_2$) were used as a positive control, while uninfected MoDCs were used as a negative control [19]. At the specified time points, cells were collected, washed three times with cold PBS with 1% penicillin and streptomycin, and then resuspended in 20 µl binding buffer. Following this, cells were stained with 1 µl of FITC Annexin V and 2 µl of 7-AAD for 15 min

at room temperature in a dark room. The percentage of apoptosis cells (FITC Annexin V$^+$/ 7-AAD$^-$ and FITC Annexin V$^+$/7-AAD$^+$) were determined by flow cytometry.

## Western blot analysis of cleaved caspase-3

MoDCs cells were infected as indicated above and then washed twice with cold PBS with 1% penicillin and streptomycin. The whole cell extracts were lysed in 50 μl of RIPA buffer (Sigma-Aldrich) and then the lysates were centrifuged at 14,000 ×g for 10 min. The supernatants were collected, and protein concentration were measured by Bradford assay (Bio-rad). 50 μg of proteins were separated by 12% Sodium dodecyl sulphate -polyacrylamide gel electrophoresis. Proteins were blotted onto polyvinylidene difluoride membranes for 45 min. The membranes were blocked with a blocking buffer (3% bovine serum albumin in Tris-buffered saline) for 1 h and then incubated with primary antibodies: polyclonal caspase-3 (dilution 1:1,000; the antibody recognizes the cleavage caspase-3) and GAPDH (dilution 1:1,000) (Cell signaling Technology) overnight. The antibody specific for cleaved caspase-3 recognizes the 17- and 19-kDa fragments but does not cross-react with full-length caspase-3 or procaspase-3 (Cell Signaling Technology, catalog number 9661). After washing, the membranes were subsequently incubated with horseradish peroxidase-conjugated secondary antibodies (diluted 1:5,000) (Thermo Fisher Scientific) for 1 h. Finally, the membranes were developed with the enhanced chemiluminescence reagents (Bio-rad) and visualized using the ChemiDoc XRS$^+$ Imaging System (Bio-Rad).

## Surface analysis of MoDCs infected with *Leptospira*

At 24 h post-infection, infected MoDCs ($2 \times 10^5$ cells) were stained with Zombie Red viability dye (BioLegend). After that, cells were stained with brilliant violet (BV) 605-labeled anti-cluster of differentiation (CD) 1a monoclonal antibody (mAb), allophycocyanin (APC)-labeled anti-CD80 mAb, FITC-labeled anti-CD83 mAb, BV785-labeled anti-CD86 mAb, APC/Cyanine7-labeled anti-CD14 mAb, and PE/Cyanine7-labeled anti-HLA-DR mAb at 4˚C for 30 min. Finally, cells were fixed in 4% paraformaldehyde. All antibodies were purchased from BioLegend Company. Cells were evaluated surface expression by flow cytometry. The graphs were plotted as the relative geometric mean fluorescence intensity (MFI) calculated as the MFI of the samples normalized to the MFI of the uninfected MoDCs (S2 Fig).

## Cytokine production of infected MoDCs and leptospirosis patients

All samples were thawed and run in the same batch to quantify the cytokine concentration. The amount of IL-6, IL-10, and IL-12p70 cytokines in the supernatants of infected MoDCs and the PGE2 production in both the supernatant of infected MoDCs and the serum of leptospirosis patients were measured using commercial ELISA kits (BioLegend) following the manufacturer's instructions. The concentration of cytokines in the samples was calculated with the standard curve and multiplied by the dilution factor.

## Co-culture of infected MoDCs and autologous naïve CD4$^+$ T cells

After 24-h infection of MoDCs with leptospires, infected MoDCs were collected and washed three times with PBS with 1% penicillin and streptomycin. The infected MoDCs were then co-cultured with autologous naïve CD4$^+$ T cells at a ratio of 1:5 in 96-well round bottom plates (Corning) in RPMI 1640 medium supplemented with 10% FBS, 1% penicillin and streptomycin, and 40 ng/ml anti-CD3 mAb.

### Intracellular cytokine staining of autologous CD4$^+$ T cells

Infected MoDCs co-cultured with autologous naïve CD4$^+$ T cells in plates were incubated at 37°C for 2 days. During the final 5 h of culture, 5 μg/ml of brefeldin A (Sigma-Aldrich) was added to block the secretion of IFN-γ, IL-4, and IL-17A. Monensin (1 μM) was added to prevent the secretion of IL-10 and FoxP3 (Biolegend). Subsequently, cells were stained with Zombie red viability dye for 30 min and stained for 20 min with BV605-labeled anti-CD1a mAb, phycoerythrin (PE) labeled anti-CD25 mAb, BV421-labeled anti-CD3 mAb, and APC/Cyanine7-labeled anti-CD4 mAb. For IFN-γ, IL-4, IL-17A, and IL-10 staining, cells were fixed in 4% paraformaldehyde and permeabilized in 1× Intracellular staining perm buffer (Biolegend) with 0.3% saponin. Cells were then stained with PerCP/Cyanine5.5-labeled anti-IFN-γ mAb, APC-labeled anti-IL-4 mAb, Alexa Flour 700-labeled anti-IL-17A mAb, and PE/Cyanine 7-labeled anti-IL-10 mAb (Biolegend). For Foxp3 staining, cells were fixed in Fix/Perm buffer and permeabilized with eBioscience buffer and stained with FITC-labeled anti-FoxP3 mAb (Thermo Fisher Scientific). IL-2 production in the supernatants were measured using LEGENDplex (Biolegend) following the manufacturer's instructions. Intracellular cytokines of CD4$^+$ T cells and IL-2 production were analyzed by flow cytometry.

### Analysis of CD4$^+$ T cell proliferation

Autologous naïve CD4$^+$ T cells were stained with 5 μM of carboxyfluorescein diacetate succinimidyl ester (CFSE) at 37°C for 20 min. The infected MoDCs were co-cultured with CFSE-labeled autologous naïve CD4$^+$ T cells at the ratio of 1:5 for 4 days. The total number of CD4$^+$ T cell proliferation was determined by flow cytometry.

### Flow cytometry analysis

After staining as described above for each experiment, flow cytometry was performed using a CytoFLEX; 3 lasers and 13 colors Fascicular (BD biosciences). The compensations were performed using single staining of antibodies in each experiment. At least 50,000–100,000 events were acquired for analysis. Unstained cells were used to calibrate the baseline photomultiplier tube voltage, presenting the background fluorescence. Data were processed using Flow-Jo software (version 10). When appropriate, fluorescence minus one sample or isotype-matched control samples was included in the staining procedure (S2 and S3 Figs).

### RNA extraction and quantification

Total RNA (n = 3/experimental group) was extracted from infected MoDCs at 4-h post-infection with TRIZOL reagent (Life Technologies Inc.) and then purified using RNeasy columns (Qiagen) according to the manufacturer's instructions. RNA samples were quantitated using a NanoDrop (ND-2000 spectrophotometer) and Qubit RNA high sensitivity (Thermo Scientific). All RNA samples had an RNA Integrity Number at least 9 before use.

### Data processing and transcriptome analysis of infected MoDCs

The libraries were processed with 20M reads per sample using the DNBseq technology platform-BGISEQ-500 (BGI Company). Genes showing a P-value < 0.05 and |Log fold-change (FC)| ≥ 1 were considered DEGs. Total DEGs in each group were normalized with total DEGs of uninfected MoDCs. Subsequently, the comparison of DEGs between pathogenic and nonpathogenic strains was defined as [DEGs of Pomona + DEGs of Clinical isolate)/2]–[(DEGs of Patoc + DEGs of Ranarum)/2]. The DEGs were subjected to Gene Ontology (GO) terms [20]

and visualized volcano plot. The RNA-seq data have been deposited in the NCBI's Sequence Read Archive under accession number PRJNA945908.

### Validation of *PTGS2* gene using reverse transcription quantitative polymerase chain reaction (RT-qPCR)

At 4-h post-infection, total RNA was extracted from infected MoDCs using a previously described method. RT-qPCR was performed using a first-strand cDNA synthesis (Bio-Rad), and then detected using QuantStudio 5 Real-time PCR system (Applied Biosystem) combined with SsoAdvanced Universl SYBR Green Supermix (Bio-Rad). The forward and reverse primer sequences of *PTGS2* were 5'-ATATGTTCTCCTGCCTACTGGAA-3' and 5'-GCCCTTCACGTTATTGCAGATG-3' [21]. The expression level of each gene was normalized to GAPDH. The $2^{-\Delta\Delta Ct}$ method was used to calculate the relative mRNA expression data [22]. All samples were compared with those of the uninfected MoDC group.

### Statistical analysis

Statistical significance was analyzed using the GraphPad Prism software (version 8, Chulalongkorn University license). Differences in parameters between groups were determined using a one-way or two-way analysis of variance (ANOVA). The Mann-Whitney test was determined for the analysis of unpaired. *P*-values < 0.05 were considered statistically significant.

## Results

### Pathogenic leptospires less activate MoDC maturation than non-pathogenic leptospires

MoDCs were infected with either pathogenic leptospires (MoDCs-P) or non-pathogenic leptospires (MoDCs-NP) at an MOI of 100, and their surface markers were assessed by flow cytometry. After 24-h infection, the expression levels of CD83 and CD80 on MoDCs-P were found to be significantly lower than those on MoDCs-NP (Fig 1A). However, no significant differences were observed in the expression levels of CD86 and HLA-DR between the two groups (Fig 1A). As expected, the expression of all surface markers was observed in LPS-treated MoDCs. These findings suggest that pathogenic leptospires can activate MoDC maturation, albeit to a lesser extent than non-pathogenic strains, potentially leading to partial impairment of MoDCs-P maturation.

Moreover, mature MoDCs can produce cytokines that consequently regulate the adaptive immunity. Therefore, the level of IL-6, IL12p-70, and IL-10 cytokines in the culture supernatants of infected MoDCs was determined using ELISA. MoDCs-P significantly produced a higher level of IL-6 but lower levels of IL-12p70 and IL-10 than MoDCs-NP (Fig 1B).

### Pathogenic leptospires induce higher MoDC apoptosis than non-pathogenic leptospires

In our preliminary study, we investigated the viability of MoDCs infected with leptospires at an MOI of 100 for 24 h, which were subsequently used to determine CD4+ T cell response. The MTS assay showed more cell death of MoDCs-P than MoDCs-NP after 24h of infection (S4 Fig). Therefore, the effect of leptospires on mature MoDCs in inducing apoptosis was determined by flow cytometry at MOIs of 10, 50, and 100 for 24 and 48 h. The infected MoDCs were stained with annexin V/7-AAD before observation (Fig 2A). Apoptotic cells were detected as early as 24 h post-infection at an MOI of 50. At 24-h of infection, pathogenic strains induced a 12% increase in MoDC apoptosis at an MOI of 100. Subsequently, the

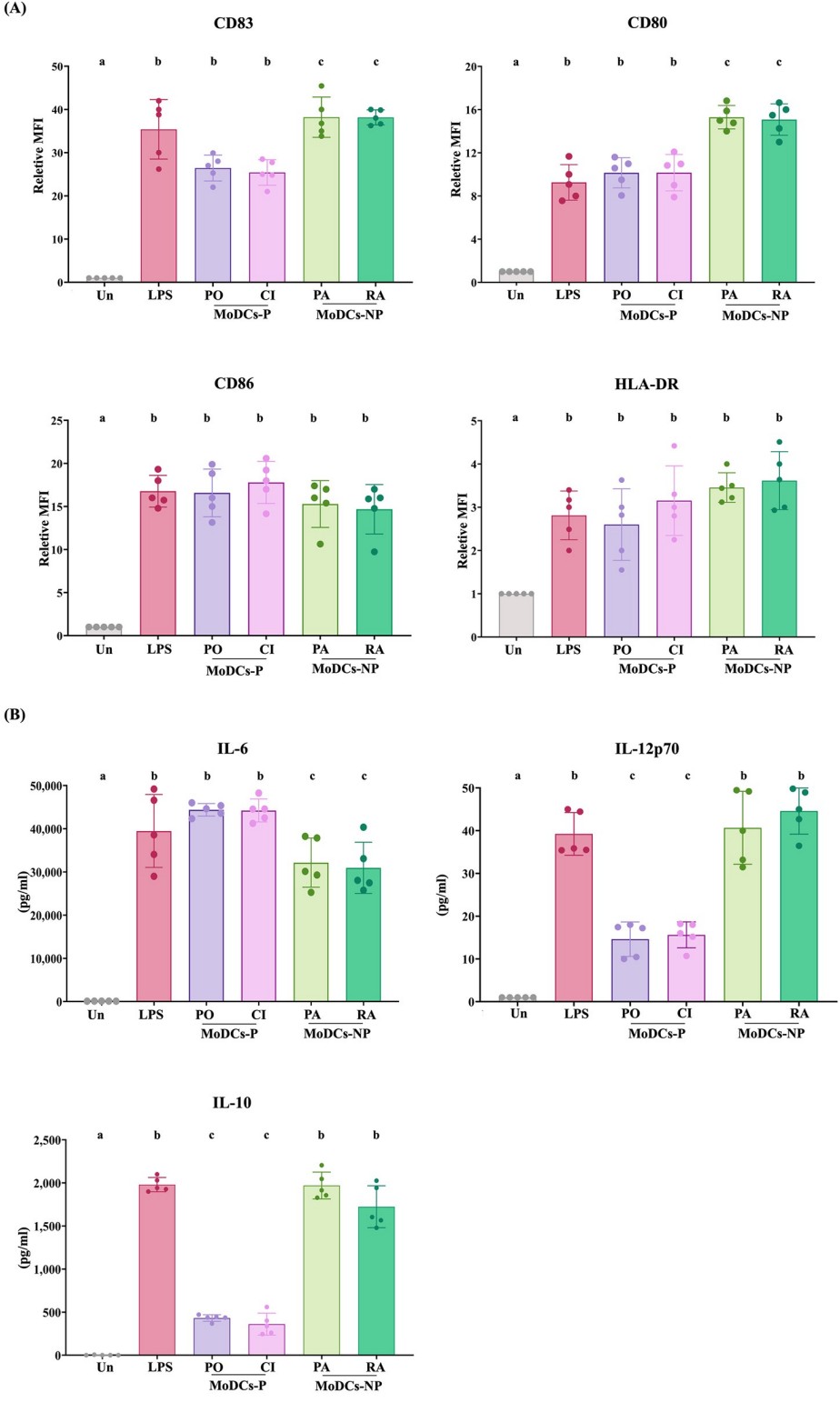

**Fig 1. Maturation and cytokine production by MoDCs-P and MoDCs-NP at 24-h post-infection.** (A) The relative geometric mean fluorescence intensities of CD83, CD80, CD86, and HLA-DR on MoDCs were assessed by flow cytometry. MoDCs were infected with pathogenic leptospires (MoDCs-P, including serovar Pomona, PO, and a clinical isolate, CI) and non-pathogenic leptospires (MoDCs-NP, including serovar Patoc, PA, and Ranarum, RA) at an MOI of 100. (B) The levels of IL-6, IL-12p70, and IL-10 in culture supernatants were measured by ELISA in both

MoDCs-P and MoDCs-NP. Uninfected MoDCs (Un) and LPS of *E. Coli* were used as the negative and positive controls, respectively. Each bar represents the mean ± SD of five donors. Statistical values were analyzed using a two-way ANOVA test. Conditions with distinct letters are significantly different ($P < 0.05$), whereas those with shared letters are not significantly different.

highest percentage of MoDC apoptosis was demonstrated at an MOI of 100 for 48 h, with approximately 80% increase in serovar Pomona and a 30% increase in clinical isolate. On the other hand, non-pathogenic leptospires at an MOI of 100 induced MoDCs apoptosis in approximately 3% and 10% of the population at 24 and 48 h, respectively (Fig 2B). However, the proportion of MoDC-NP apoptosis was not significantly different from uninfected MoDCs. Additionally, MoDCs treated with 100 μM $H_2O_2$, as a positive control, revealed approximately 25% and 60% apoptosis after 24 and 48 h, respectively. Therefore, these finding demonstrated that pathogenic leptospires induced MoDC apoptosis *in vitro* in a dose- and time-dependent manner.

Subsequently, detection of cleaved caspases-3 was conducted to support the occurrence of apoptosis and its involvement in leptospires-induced MoDC apoptosis. Cleaved caspase-3 is an active form of caspase-3 that plays a role in executing the process of apoptosis. It also serves as a marker for detecting apoptotic cells [23]. Immunoblot analysis using an antibody specific for cleaved caspase-3, which does not cross-react with procaspase-3, revealed the production of caspase-3 cleavage fragments at approximately 17 and 19 kDa. At an MOI of 100, pathogenic leptospires exhibited a higher intensity of cleaved caspase-3 than non-pathogenic leptospires at 48 h post-infection (Fig 2C). However, the band intensity of cleaved caspase-3 production was similar in both pathogenic and non-pathogenic leptospires at 24 h post-infection (S5 Fig). Taken together, these findings suggest that pathogenic leptospires induce greater MoDC apoptosis via caspase-3 activation when compared to non-pathogenic strains.

## MoDCs-P induce Treg cells but fail to promote CD4$^+$ T cell proliferation

To determine the effect of MoDCs-P on the initiation of adaptive immune responses, the activation, polarization, and proliferation of primary human CD4$^+$ T cells were observed after co-culture with the infected MoDCs. The T-cell activation-induced markers and cytokines were measured by flow cytometry after 48 h co-culture. The analysis revealed an increase in the CD25 surface marker (S6 Fig), suggesting that both MoDCs-P and MoDCs-NP could induce CD4$^+$ T cell activation. The expression of different cytokines and markers was selected to reflect the polarization state of activated CD4$^+$ T cells as follows: IFN-γ for Th1 cells, IL-4 for Th2 cells, IL-17A for Th17 cells; and FoxP3 and IL-10 for Treg cells. The results demonstrated that the percentage of CD4$^+$ T cells expressing IL-10 and FoxP3 were significantly higher in MoDCs-P than MoDCs-NP (Fig 3A and 3B). In contrast, MoDCs-NP led to an increase in IFN-γ-expressing CD4$^+$ T cells (Fig 3C). The expression of IL-4 and IL-17A (Fig 3D and 3E), and the secretion of IL-2 (S6 Fig) showed no differences in both groups. These results indicate that MoDCs-P enhance induction of regulatory T cells, while MoDCs-NP induce Th1 cells.

Moreover, the effect of MoDCs infected with leptospires on CD4$^+$ T cell proliferation was investigated. The CFSE-labeled autologous naïve CD4$^+$ T cells were co-cultured with the infected MoDCs for 4 days and CD4$^+$ T cell proliferation was measured by flow cytometry (Fig 4A). The result demonstrated that MoDCs-P significantly induced lower proliferation of CD4$^+$ T cells than MoDCs-NP (Fig 4B). Taken together, our findings demonstrate that the MoDCs-P may induce Treg cell polarization while failing to induce CD4$^+$ T cell proliferation.

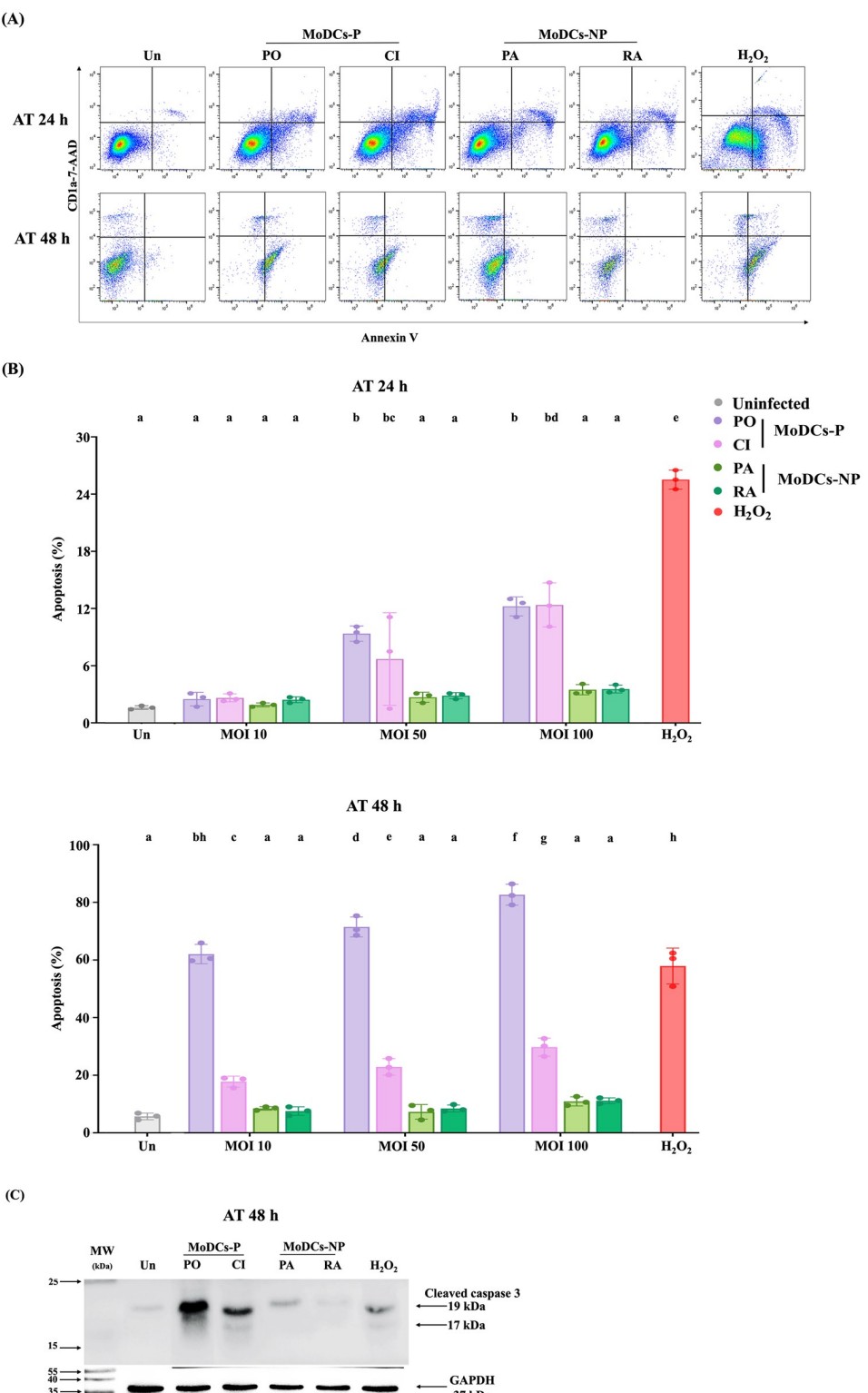

**Fig 2. Apoptosis induction in MoDCs-P and MoDCs-NP.** (A) Representative scatter plots for MoDC apoptosis induced by pathogenic leptospires (MoDCs-P, including serovar Pomona, PO, and a clinical isolate, CI) and non-pathogenic leptospires (MoDCs-NP, including serovar Patoc, PA, and Ranarum, RA) at an MOI of 100 for 24 and 48 h. The data shown are representative of three independent experiments. The infected MoDCs were stained with annexin V/7-AAD, and the florescence signals were quantified by flow cytometry. (B) The percentage of apoptosis was

determined by flow cytometry at MOIs of 10, 50, and 100 for 24 and 48 h. (C) Representative Western blot analyses were conducted using samples from three donors to determine cleavage of caspase-3 in MoDCs-P and MoDCs-NP following infection at an MOI of 100 for 48 h. GAPDH served as an internal control. MoDCs treated with 100 μM $H_2O_2$ were used as a positive control, and an uninfected group (Un) used as a negative control. Each bar represents the mean ± SD of three donors. Statistical values were analyzed using a two-way ANOVA test. Conditions with distinct letters are significantly different ($P < 0.05$), whereas those with shared letters are not significantly different.

On the other hand, non-pathogenic strains demonstrated the ability to stimulate greater Th1 cell activation as well as enhanced CD4+ T cell proliferation.

## Pathogenic leptospires induce the transcription of *PTGS2* gene in MoDCs

We found that pathogenic strains partially impaired MoDC responses, which potentially led to Treg polarization. To investigate this further, the transcriptomic profiles of MoDCs were analyzed at 4 h post-infection. After normalization using uninfected MoDCs as a control within each group, we identified a total of 1,316 differentially expressed genes between MoDCs-P and MoDCs-NP. Among these, 123 genes were expressed at significantly higher levels, and 1,193 genes were expressed at significantly lower levels, in comparison to MoDCs-NP. The top 5 differentially expressed genes (DEGs) were both indicated on a volcano plot (Fig 5A) and shown in Table 1. The gene encoding prostaglandin-endoperoxide synthase 2 (*PTGS2*), also known as cyclooxygenase-2 (COX-2), emerged as a compelling target for further investigation because its PGE2 product has been previously shown to reduce T cell proliferation and promote the activation of Tregs [24], observations that are consistent with our findings in MoDCs-P. Therefore, the *PTGS2* gene was selected for validation of its expression level using relative RT-qPCR. The results showed that the expression levels of *PTGS2* in MoDCs-P were upregulated more than 50-fold compared to uninfected cells. These fold changes were significantly higher than those observed in MoDCs-NP (Fig 5B).

## Prostaglandin E2 secreted by MoDCs infected with leptospires and in sera of patients with acute leptospirosis

The level of PGE2 was measured in the culture supernatants of infected MoDCs as well as in the sera of patients in the acute phase of leptospirosis, using ELISA to confirm its relevance to leptospiral infection. The results demonstrated that, at 24 h post-infection, MoDCs-P produced significantly higher levels of PGE2 compared to both MoDCs-NP and uninfected MoDCs (Fig 6A). Moreover, the levels of PGE2 in the sera of patients with acute leptospirosis were significantly elevated compared to those in healthy donors (Fig 6B). These findings indicate that pathogenic leptospiral infection induces the production of PGE2 both *in vitro* and *in vivo*.

## Discussion

Dendritic cells are key antigen-presenting cells that initiate antigen-specific immune responses against microbial infections. DCs can modulate the adaptive immune response by polarizing lymphocytes, thereby affecting the outcome of bacterial infections [12]. Nevertheless, the responses of MoDCs in concert with CD4+ T cells against leptospiral infection in humans has not yet been investigated, thereby establishing the objective of this study.

In this study, we compared the responses of human MoDCs infected with live pathogenic and non-pathogenic leptospires as well as their interplay with CD4+ T cells. Given that viable leptospires have shown greater potency in activating immune cells compared to their heat-

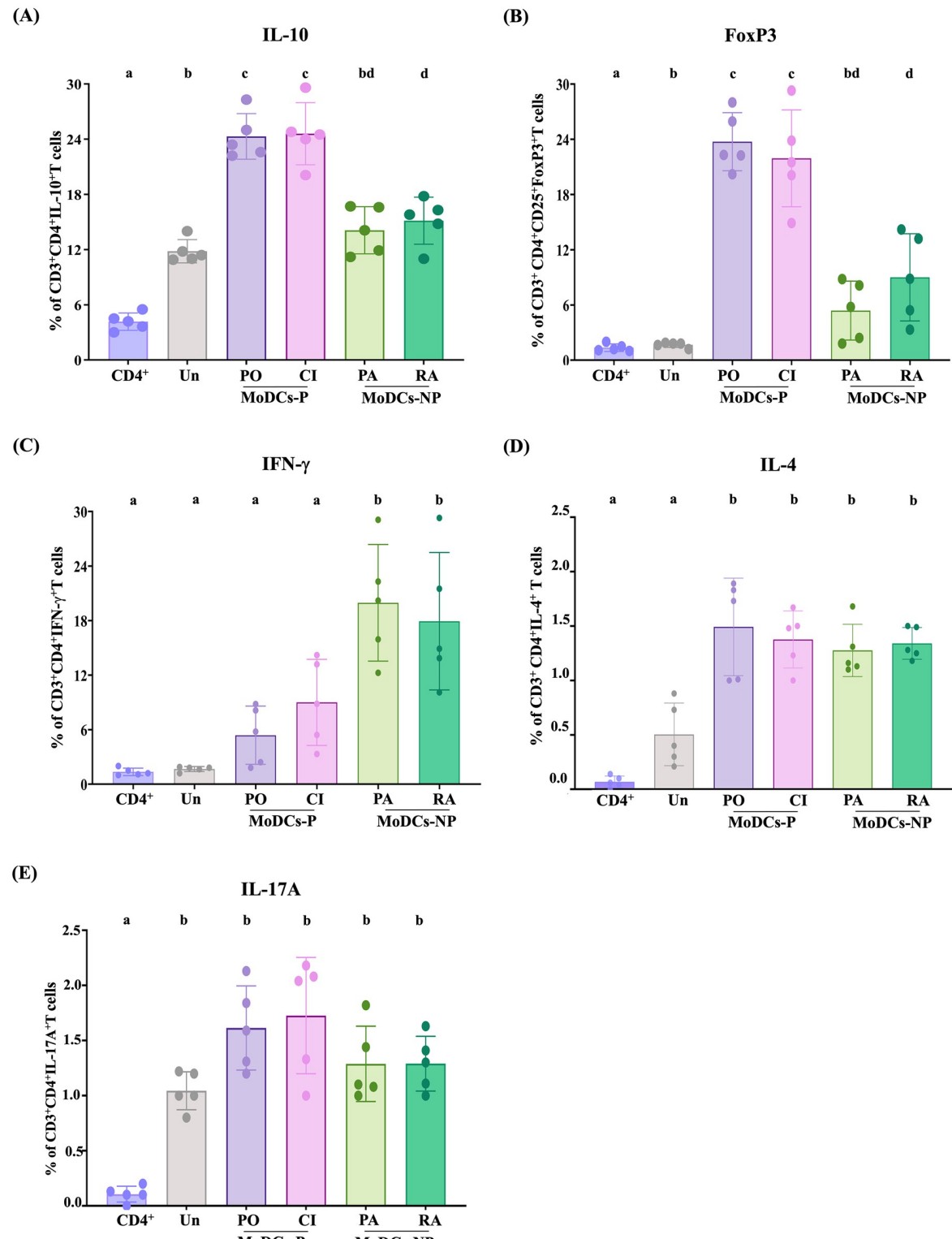

**Fig 3. Intracellular cytokine production in autologous CD4⁺ T cells activated by infected MoDCs after 2 days of co-culture.**
Expression of (A) IL-10, (B) FoxP3, (C) IFN-γ, (D) IL-4, and (E) IL-17A in CD4⁺ T cells activated by MoDCs infected with pathogenic leptospires (MoDCs-P, including Pomona, PO, and a clinical isolate, CI) and non-pathogenic leptospires (MoDCs-NP, including serovar Patoc, PA, and Ranarum, RA). After 24 h of infecting MoDCs with leptospires, the infected MoDCs were co-cultured with CD4⁺ T cells at a ratio of 1:5 for 48 h. Uninfected MoDCs (Un) and CD4⁺ T cells were used as negative controls. Each bar represents the mean ± SD of five

donors. Conditions with distinct letters are significantly different ($P < 0.05$), whereas those with shared letters are not significantly different.

killed counterparts [25], we infected MoDCs with live, low-passage leptospires originally isolated from clinical samples. To account for potential variations between strains, we ensured result consistency across strains within each *Leptospira* group, thereby validating that the differences in MoDCs-P and MoDCs-NP, as well as corresponding T cell responses, were not due to strain-specific effects.

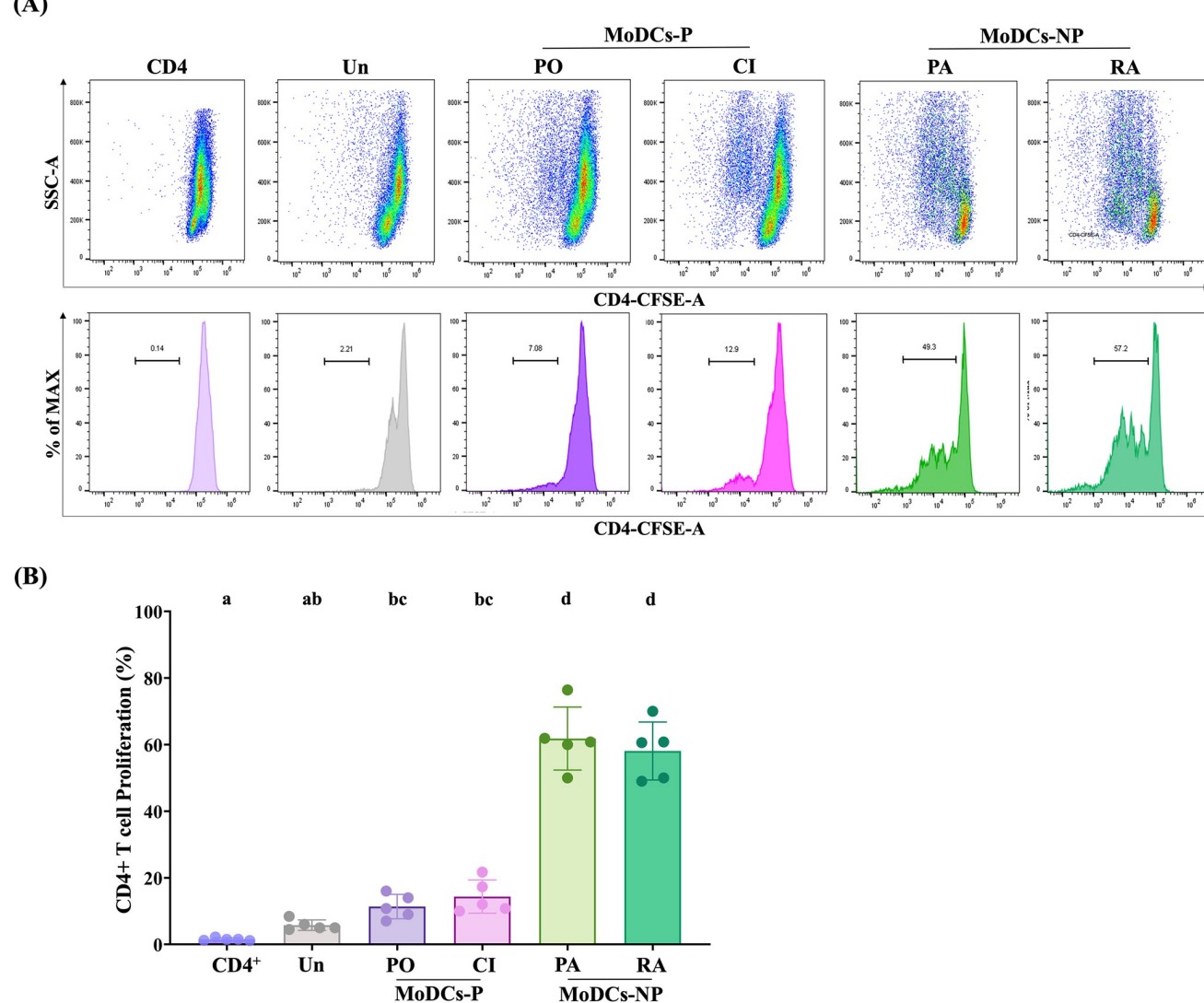

**Fig 4. Induction of autologous CD4+ T cell proliferation by infected MoDCs after 4 days of co-culture.** (A) Representative flow cytometry scatter plots and histograms demonstrating CFSE-labeled CD4+ T cell proliferation. At 24 h post-infection, MoDCs infected with pathogenic leptospires (MoDCs-P, including serovar Pomona, PO, and a clinical isolate, CI) and non-pathogenic leptospires (MoDCs-NP, including serovar Patoc, PA, and Ranarum, RA) were co-cultured with CFSE-labeled CD4+ T cells at a ratio of 1:5 for 4 days. (B) The percentage of CD4+ T cell proliferation was determined by flow cytometric analysis. Uninfected MoDCs (Un) and autologous CD4+ T cells were used as negative controls. Each bar represents the mean ± SD of five donors. Conditions with distinct letters are significantly different ($P < 0.05$), whereas those with shared letters are not significantly different.

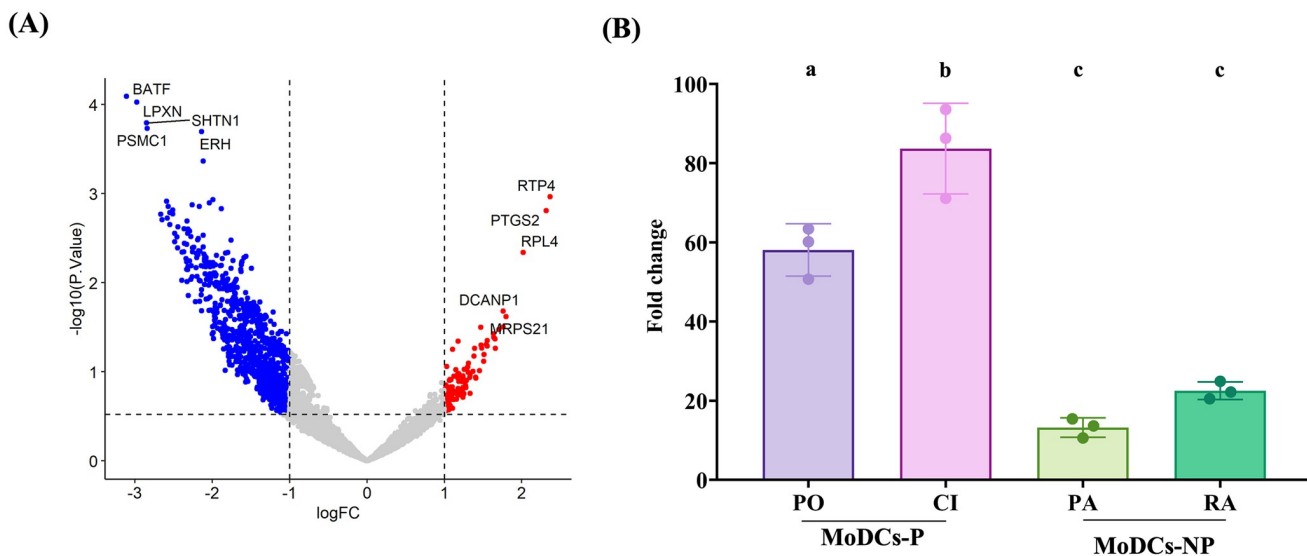

**Fig 5. Differentially expressed genes (DEGs) in MoDCs-P and MODCs-NP at 4 h post-infection.** (A) Volcano plot of RNA-seq data. MoDCs were infected with pathogenic leptospires (MoDCs-P, including serovar Pomona, PO, and a clinical isolate, CI) and non-pathogenic leptospires (MoDCs-NP, including serovar Patoc, PA, and Ranarum, RA) at 4 h post-infection. The dotted vertical lines bound the minimal fold-change for the most differentially expressed genes. Red dots represent upregulated genes, and blue dots represent downregulated genes (adjusted *P*-value < 0.3, and fold change > 1). The data represent results from three donors. (B) Relative RT-qPCR analysis of *PTGS2* expression in MoDCs infected with leptospires for 4 h, calculated relative to uninfected MoDCs (Un). Each bar represents the mean ± SD of three donors. Conditions with distinct letters are significantly different (*P* < 0.05), whereas those with shared letters are not significantly different.

Our findings reveal that pathogenic leptospires partially impair MoDC maturation and skew MoDCs-CD4[+] T cell interactions toward Treg cells, thereby failing to stimulate CD4[+] T cell proliferation. In line with previous work [14], MoDCs-P exhibited lower expression levels of CD83 and CD80 compared to MoDCs-NP. Although CD86 and HLA-DR did not show significant differences, the overall expression profile of surface markers suggests impaired maturation of MoDCs-P relative to MoDCs-NP. Beyond serving as a prominent marker for mature DCs, CD83 also play a role in regulating T cell responses and maintaining immune system balance [26]. Previous research has shown that downregulated CD83 expression on human DCs resulted in less potent induction of T cell proliferation and reduced IFN-γ secretion by T cells [27].

**Table 1. The top 5 upregulated and downregulated genes in MoDCs-P based on transcriptomic analysis at 4 h post-infection.**

| Regulation | Genes | Log fold change | Adjusted *P*-value | *P*-value |
|---|---|---|---|---|
| Up | Receptor transporter protein 4 (*RTP4*) | 2.36 | 0.30 | 0.001 |
| | Prostaglandin endoperoxide synthase 2 (*PTGS2*) | 2.31 | 0.30 | 0.001 |
| | Ribosomal protein L4 (*RPL4*) | 2.02 | 0.30 | 0.004 |
| | Dendritic cell associated nuclear protein (*DCANP1*) | 1.76 | 0.44 | 0.020 |
| | Mitochondrial ribosomal protein S21 (*MRPS21*) | 1.79 | 0.44 | 0.02 |
| Down | Basic leucine zipper ATF-like transcription factor (*BATF*) | -3.11 | 0.19 | 8.114 |
| | Leupaxin (*LPXN*) | -2.94 | 0.19 | 9.39 |
| | Shootin 1 (*SHTN1*) | -2.84 | 0.19 | 0.001 |
| | Proteasome 26s subunit ATPase 1 (*PSMC1*) | -2.84 | 0.19 | 0.001 |
| | ERH MRNA splicing and mitosis factor (*ERH*) | -2.13 | 0.19 | 0.001 |

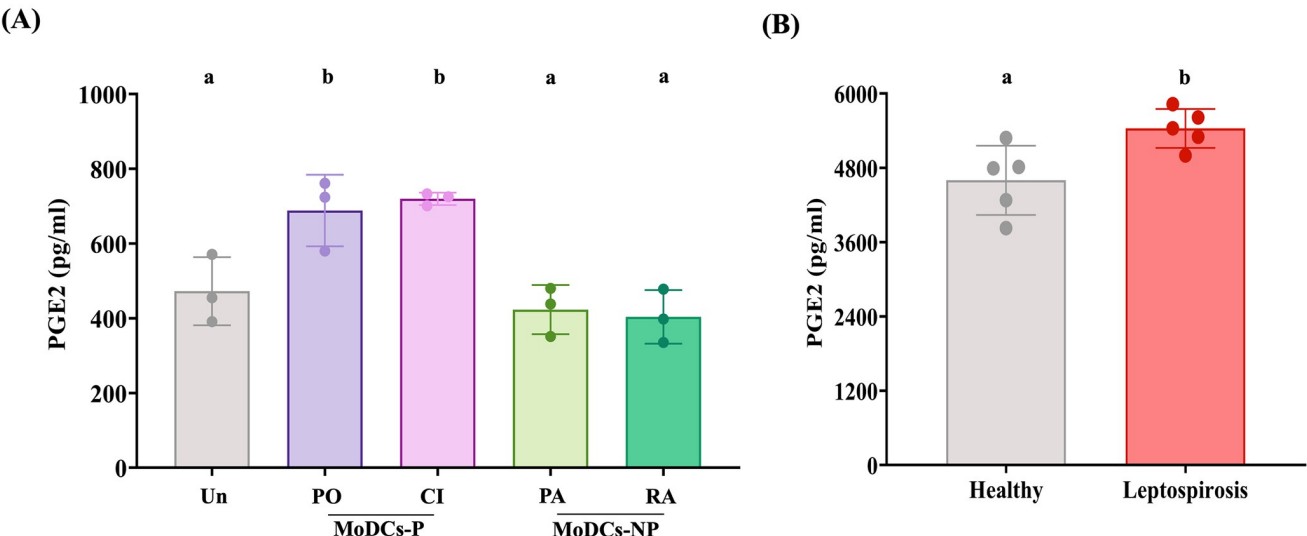

**Fig 6. PGE2 levels in culture supernatants of infected MoDCs and sera of patients with acute leptospirosis.** (A) PGE2 levels were measured in MoDCs infected with pathogenic leptospires (MoDCs-P, including serovar Pomona, PO, and a clinical isolate, CI) and non-pathogenic leptospires (MoDCs-NP, including serovar Patoc, PA, and Ranarum, RA) at 24 h post-infection. Uninfected MoDCs (Un) were used as negative controls. (B) Comparison of PGE2 levels in sera of patients with acute leptospirosis with those of healthy controls. PGE2 levels were quantified using ELISA. Each bar represents the mean ± SD from three independent donors for culture supernatant measurements and five independent donors for patient sera measurements. Statistical values were analyzed using the Mann-Whitney test. Conditions with distinct letters are significantly different ($P < 0.05$), whereas those with shared letters are not significantly different.

Furthermore, strong co-stimulatory signals from CD80 and CD86 are essential for the activation of naïve T cells *in vitro* [28]. Although CD80 and CD86 share the same receptors (CD28 or CTLA-4) on T cells, they exhibit distinct binding affinities and kinetics, leading to distinct functional outcomes. Our results demonstrated that MoDCs-P exhibited lower CD80 expression compared to MoDCs-NP, but CD86 expression levels were not significantly different. This suggests that the interaction involving CD80 may potentially affect the binding force between CD28 or CTLA-4 receptors on CD4$^+$ T cells. These findings align with previous studies that showed inhibiting CD80 on mouse dendritic cells reduced the interaction forces between DCs and T cells, consequently impairing T cell activation and reducing IFN-γ secretion [28]. Given that our study focused on a single time point (24 h post-infection), it is worth noting that the abundance and differential expression of MoDC surface markers may dynamically change upon exposure to inflammatory stimuli, including leptospiral infection. Additional research is needed to elucidate the roles and mechanisms of CD80 and CD83 in MoDC responses to pathogenic leptospires.

Our study also examined cytokine production but was limited to IL-12p70, IL-6, and IL-10. Despite this limitation, we observed higher levels of IL-10 and IL-12p70 secretion in MoDCs-NP compared to MoDCs-P. These findings align with previous studies that have demonstrated greater IL-10 production in MoDCs infected with saprophytic leptospires (serovar Patoc) than those infected with the pathogenic leptospires strain Verdun [14]. Furthermore, MoDCs-NP exhibited higher levels of IL-12p70 production than MoDCs-P, which may potentially indicate an induction toward Th1 cell responses. However, further studies are required to elucidate the role of cytokines in MoDC responses to leptospiral infection.

At the transcriptional level, MoDCs-P displayed significant upregulation of the *PTGS2* gene and increased secretion of its product, PGE2, both in culture supernatants of MoDCs-P and in the sera of patients with acute leptospirosis when compared to uninfected control cells and

healthy donor, respectively. PGE2 is a lipid mediator synthesized through the conversion of arachidonic acid by PTGS2 enzymes [29,30]. The PTGS2/PGE2 pathway is known to regulate inflammatory responses and is also involved in apoptosis and immunosuppression [31]. These prostaglandins may exert immunosuppressive effects by reducing T cell proliferation and promoting the activation and expansion of Tregs. These observations are in line with our findings, as we observed decreased T cell proliferation and induction of Tregs in response to *PTGS2*-expressing MoDCs-P. Moreover, previous studies demonstrated an upregulation of *PTGS2* expression in the blood and target organs of hamsters infected with pathogenic leptospires [32,33]. Additionally, prior research has shown that the combination of PGE2 and IL-10 production can convert naïve CD4$^+$ T cells into Tregs [34–36]. These results suggest that the induction of Tregs by MoDCs-P may be related to the activation of *PTGS2* gene. Furthermore, previous research indicates that PGE2 is induced in and released by dying cells, functioning as an inhibitory damage-associated molecular pattern (iDAMP) that suppresses immune responses and inflammation [37]. Therefore, it is possible that the observed higher levels of PGE2 secreted by MoDCs-P could result from increased apoptosis in these cells compared to MoDCs-NP. This elevated PGE2 level could then act as an iDAMP, thereby potentially impairing the responses of CD4$^+$ T cells. However, further evaluation is needed to fully elucidate the role of the PTGS2/PGE2 pathway in both infected and apoptotic MoDC cell responses to leptospiral infection.

Apoptosis of dendritic cells plays a crucial role in regulating the immune system by controlling antigen presentation to T cells. Alterations in dendritic cell death can significantly influence the immune response directed towards antigens, as well as impact processes like inflammation and the maintenance of immune tolerance [38]. Interestingly, our results suggested that pathogenic leptospires induced greater apoptosis in MoDCs than non-pathogenic leptospires, potentially via caspase-3 activation. We observed a statistically significant increase in MoDC apoptosis, especially in those infected with pathogenic *L. interrogans* for 48 h. These apoptotic MoDCs could be less effective as antigen-presenting cells, potentially impairing the activation of CD4$^+$ T cells. Notably, for the assessment of the CD4$^+$ T cell responses, we used MoDCs infected with leptospires at an MOI of 100 for 24 h. At this time point, cell viability was observed to be at least 80% in both MoDCs-P and MoDCs-NP, likely providing a sufficient number of MoDCs for CD4$^+$ T cell activation. Other pathogens, such as *Streptococcus pneumoniae* and *Listeria monocytogenes*, have similarly developed strategies to induce apoptosis in DCs as a means of evading the immune response, consequently contributing to diseases progression [39,40]. Previous studies have shown that pathogenic leptospires induced apoptosis in guinea pig hepatocytes and a monocyte-macrophage-like cell line through caspase-8 and caspase-3-dependent pathways [41,42]. Pathogenic leptospires have also been reported to induce apoptosis in myeloid neutrophils and monocytes [43]. Thus, our result suggests that inducing MoDC apoptosis may be a strategy used by pathogenic leptospires to evade immune responses. However, previous research regarding other types of cell death induced by pathogenic leptospires has not been reported in human DCs. Necroptosis has been shown to be involved in the innate immune response and contribute to persistent inflammation during acute leptospirosis in C3H/HeJ mice [43]. A previous study showed that pathogenic leptospires did not induce pyroptosis in murine, bovine, human, and hamster macrophages [44]. The limitations of the apoptosis experiment in the present study include a small sample size and the lack of an investigation into other cell death mechanisms. Increasing the sample size may strengthen the conclusions regarding the ability of pathogenic leptospires to induce apoptosis as a means to evade the CD4$^+$ T cells response. To the authors' knowledge, this is the first report to study apoptosis in human MoDCs induced by pathogenic leptospires through caspase-3 activation. Further research is necessary to gain a more comprehensive understanding.

Another interesting finding in this study is that non-pathogenic strains appeared to enhance T cell proliferation and promote the production of IFN-γ-producing CD4$^+$ T cells, indicative of a Th1 response. In the context of natural human infections, there is no current evidence whether non-pathogenic strains are cleared by innate immune responses or whether the induction of adaptive immunity is required. Although saprophytic leptospires are sensitive to the alternative pathway of complement-mediated killing *in vitro* [9], it remains unclear whether this mechanism is sufficient to eradicate non-pathogenic leptospires *in vivo*. A recent study demonstrated that live saprophytic leptospires could exit human macrophage without causing cell lysis and were resistant to classical macrophage microbicidal mechanisms such as phagocytosis, autophagy, toll-like receptor-mediated reactive oxygen (ROS) species, and reactive nitrogen species (RNS) production [45]. It is possible that, after exiting the macrophages, other immune strategies including Th1 response may contribute to bacterial clearance. A combination of innate and adaptive immune mechanisms might be required to eliminate non-pathogenic leptospires from the host. Therefore, our findings suggest, but do not definitely establish, that the induction of a Th1 response may be involved in the clearance of non-pathogenic leptospires. Additional studies are needed to elucidate the mechanisms responsible for their elimination.

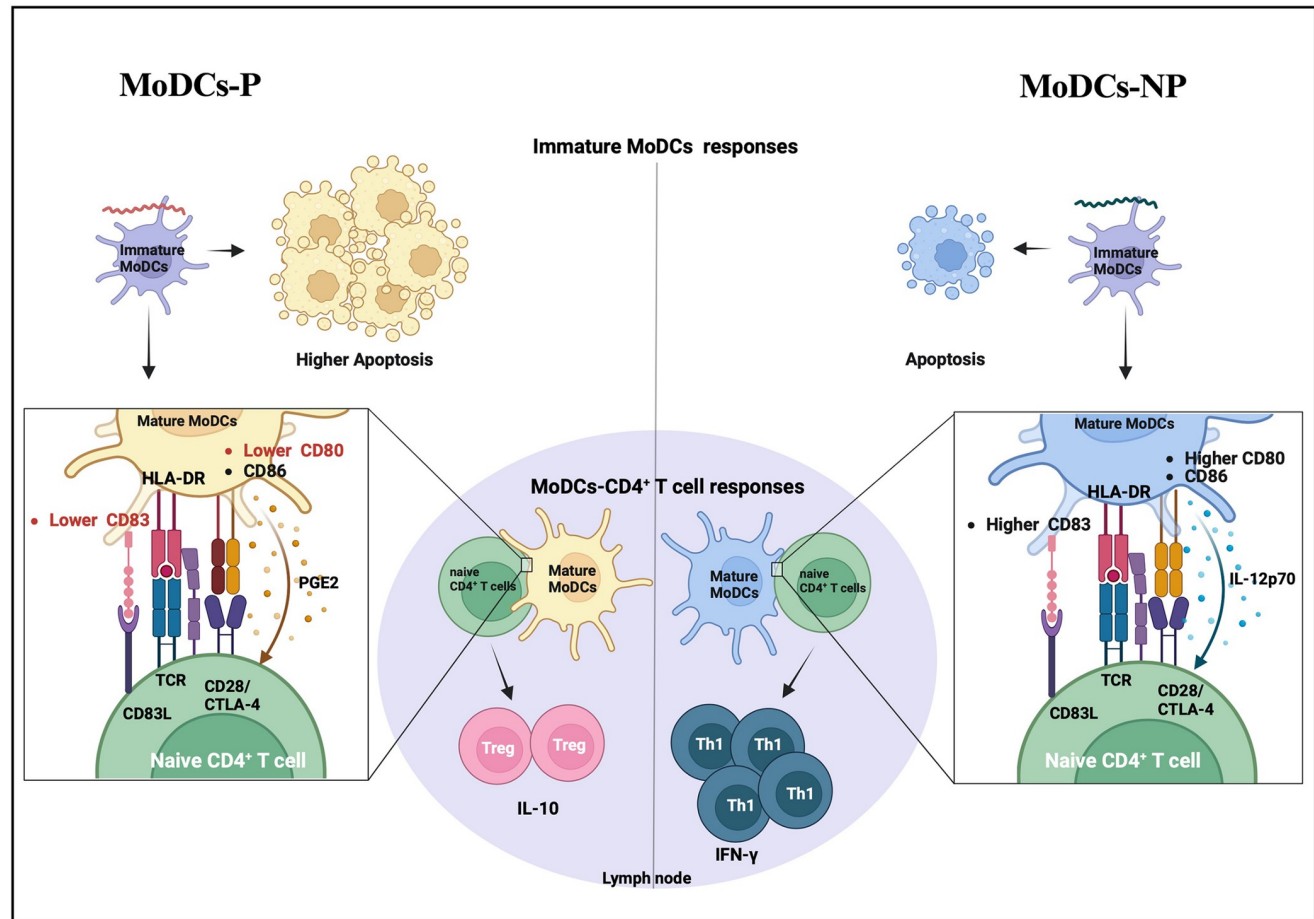

**Fig 7. Schematic representation of MoDCs-CD4$^+$ T cell responses to pathogenic and non-pathogenic leptospires.** MoDCs-P represent MoDCs infected with pathogenic leptospires, while MoDCs-NP represent MoDCs infected non-pathogenic leptospires. The description is stated in the main text. The illustration is based on findings from this study and omits details not directly relevant to the findings. Figure created using BioRender.com.

In conclusion, this study expands our understanding of the potential roles of DCs and CD4[+] T cells in response to leptospiral infection, as illustrated in Fig 7. Pathogenic leptospires appear to modulate the interplay of MoDCs-CD4[+] T cells toward Treg cell activation while failing to stimulate CD4[+] T cell proliferation, in part by partially impairing MoDC maturation and upregulating the PTGS2/PGE2 pathway. In contrast, non-pathogenic leptospires may enhance MoDC maturation and promote a Th1 response, potentially facilitating their elimination in concert with other immune mechanisms. These findings provide novel insights into the pathogenesis of leptospirosis and highlight a new approach to develop vaccines that modulate DC responses, ultimately improving the host immunity to clear the pathogen.

## Supporting information

**S1 Fig. Gating strategies for analyzing cell purity.**
(TIF)

**S2 Fig. The surface expression of CD83, CD80, CD86, and HLA-DR of infected MoDCs.**
(TIF)

**S3 Fig. Isotype controls for quantifying the gating of intracellular cytokine production in CD4[+] T cells.**
(TIF)

**S4 Fig. Percentage of viable MoDCs infected with leptospires assessed by MTS assay.**
(TIF)

**S5 Fig. Representative Western blot analyses of cleaved caspases-3 in MoDCs-P and MoDCs-NP following infection at an MOI of 100 for 24 h.**
(TIF)

**S6 Fig. The percentage of CD3[+]CD4[+]CD25[+] T cells and IL-2 production after co-culture with infected MoDCs.**
(TIF)

## Acknowledgments

We would like to thank Ben Alder for kindly providing *L. interrogans* serovar Pomona, and also thank the Melioidosis Research Center, Khon Kaen University, Thailand, for maintaining this strain. We also thank the doctors and staff at Hat Yai Hospital and Surin Hospital for their kind collaborations in providing patient samples.

## Author Contributions

**Conceptualization:** Pratomporn Krangvichian, Kanitha Patarakul.

**Data curation:** Pratomporn Krangvichian, Teerasit Techawiwattanaboon.

**Formal analysis:** Pratomporn Krangvichian, Teerasit Techawiwattanaboon, Tanapat Palaga, Patcharee Ritprajak, Patipark Kueanjinda.

**Funding acquisition:** Pratomporn Krangvichian, Kanitha Patarakul.

**Investigation:** Pratomporn Krangvichian, Teerasit Techawiwattanaboon.

**Methodology:** Pratomporn Krangvichian, Tanapat Palaga, Patcharee Ritprajak, Chamraj Kaewraemruaen, Kanitha Patarakul.

**Project administration:** Kanitha Patarakul.

**Resources:** Kanitha Patarakul.

**Supervision:** Tanapat Palaga, Patcharee Ritprajak, Kanitha Patarakul.

**Validation:** Pratomporn Krangvichian, Teerasit Techawiwattanaboon, Tanapat Palaga, Patchaaree Ritprajak, Patipark Kueanjinda, Kanitha Patarakul.

**Visualization:** Teerasit Techawiwattanaboon, Patipark Kueanjinda.

**Writing – original draft:** Pratomporn Krangvichian.

**Writing – review & editing:** Teerasit Techawiwattanaboon, Tanapat Palaga, Patcharee Ritprajak, Patipark Kueanjinda, Kanitha Patarakul.

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
