## [Decision Letter · Decision Letter 0]

15 May 2023

Dear Dr. Patarakul,

Thank you very much for submitting your manuscript "Impaired functions of human monocyte-derived dendritic cells and induction of regulatory T cells: a potential mechanism of immune evasion of pathogenic Leptospira" for consideration at PLOS Neglected Tropical Diseases. As with all papers reviewed by the journal, your manuscript was reviewed by members of the editorial board and by several independent reviewers. In light of the reviews (below this email), we would like to invite the resubmission of a significantly-revised version that takes into account the reviewers' comments. 

While all the reviewers acknowledged the importance and innovative aspects of your work, they raised concerns with writing clarity, data interpretation and methodological approaches of your manuscript. The reviewers requested additional experiments to validate some of your conclusions, particularly experiments related to phagocytosis and apoptosis studies. Please, carefully address all the comments listed by the reviewers before submitting a revised version of your manuscript and let the journal know if you will need additional time to prepare your revised manuscript.

We cannot make any decision about publication until we have seen the revised manuscript and your response to the reviewers' comments. Your revised manuscript is also likely to be sent to reviewers for further evaluation.

Sincerely,

Andre Alex Grassmann, PhD

Guest Editor

Elsio Wunder Jr

Section Editor

While all the reviewers acknowledged the importance and innovative aspects of your work, they raised concerns with writing clarity, data interpretation and methodological approaches of your manuscript. The reviewers requested additional experiments to validate some of your conclusions, particularly experiments related to phagocytosis and apoptosis studies. Please, carefully address all the comments listed by the reviewers before submitting a revised version of your manuscript and let the journal know if you will need additional time to prepare your revised manuscript.

Reviewer's Responses to Questions

**Key Review Criteria Required for Acceptance?**

**Methods**

-Are the objectives of the study clearly articulated with a clear testable hypothesis stated?

-Is the study design appropriate to address the stated objectives?

-Is the population clearly described and appropriate for the hypothesis being tested?

-Is the sample size sufficient to ensure adequate power to address the hypothesis being tested?

-Were correct statistical analysis used to support conclusions?

-Are there concerns about ethical or regulatory requirements being met?

Reviewer #1: Please check Summary and General Comments

Reviewer #2: The review comments are finalized in uploaded documents.

Reviewer #3: Not quite (see below)

**Results**

-Does the analysis presented match the analysis plan?

-Are the results clearly and completely presented?

-Are the figures (Tables, Images) of sufficient quality for clarity?

Reviewer #1: Please check Summary and General Comments

Reviewer #2: The review comments are finalized in uploaded documents.

Reviewer #3: See below

**Conclusions**

-Are the conclusions supported by the data presented?

-Are the limitations of analysis clearly described?

-Do the authors discuss how these data can be helpful to advance our understanding of the topic under study?

-Is public health relevance addressed?

Reviewer #1: Please check Summary and General Comments

Reviewer #2: The review comments are finalized in uploaded documents.

Reviewer #3: see below

**Editorial and Data Presentation Modifications?**

Reviewer #1: Please check Summary and General Comments

Reviewer #2: The review comments are finalized in uploaded documents.

Reviewer #3: Yes, this would be required (see below)

**Summary and General Comments**

Reviewer #1: Overall, 

The in vitro study proposes some insights into the mechanism of how pathogenic vs non-pathogenic strains of leptospires evade the host immune system. Briefly, the authors propose that MoDCs are rather activated/maturated by non-pathogenic than by pathogenic strains of leptospires. Therefore, while NP strains induce a pro-resolution response, pathogenic strains would dampen immune response by inducing MoDCs apoptosis, and the secretion of regulatory cytokines by CD4+ T cells. Interestingly, authors have found PTGS2 as a potential key regulator in MoDCs for leptospire strains' susceptibility. Although very important data are being presented in the current study, major improvements in the manuscript are necessary for a version to be published. 

Minor comments

-The letters system adopted by the authors to indicate statistical difference is not obvious.

-Authors need to standardize that cells were “INFECTED” and not “STIMULATED”; both terms are being used to indicate the same thing.

-Authors named conditions in the text as MoDCs-P and MoDCs-NP. However, the figures do not follow the same nomenclature. Standardizing the condition’s names on both text and figures would help for faster comprehension.

Major comments

Lines 299-312

Authors state that pathogenic strains are mostly visible and attached outside MoDCs compared to non-pathogenic strains, located “intracellularly”. I cannot see the same only by visualizing the presented images. Apparently, figure 3A indicate: PO=PA and CI=RA. 

1) Fluorescence quantitative analysis of different z-stack images obtained in confocal microscopy is necessary for supporting the author’s statement. 

2) Additionally, CFU phagocytosis assay, using antibiotics a few minutes after infection (to eliminate extracellular bacteria) would be helpful to clarify if bacteria are being internalized or not.

Lines 333-342

I'm wondering why the authors didn't present figures S4-S5 together in Fig4. Cell death analysis needs different approaches/analyses to be undoubted. However, there are some concerns, regarding cell death, that need to be further explored by the authors. Both P and NP strains induced a very low rate of apoptosis. Some additional controls and conditions are required for authors to highlight the biological relevance of apoptosis induced by leptospire strains. 

1) An apoptotic positive control is missing in these experiments, e.g. staurosporine, Viscum album, etc.

2) Also, analysis at different time points after infection is necessary to support the authors' statement that pathogenic strains induce apoptosis in MoDCs. By the way, the time-point after infection is not clear in the text/legend.

3) Likewise, different MOIs must be used in these experiments to help authors support their conclusion. Again, no mention of MOI used in these experiments.

Lines 381-388

It is not clear for how long MoDCs were prior infected before co-culture with CD4+ T cells. Figure S7 should be placed within Fig6 for a more accurate illustration of the events taking place.

Lines 400-414

In the results section, the authors highlighted that PTGS2 was the most intriguing DEG but didn’t explain Why. Although PTGS2’s role is further explored in the discussion, it would be important for readers a brief paragraph or sentence explaining why authors end up pondering PTGS2's importance in the current study. Also, Fig7A is poorly explored. It does not inform much other than that authors have chosen the PTGS2 (again, for a non-mentioned reason). Still regarding PTGS2, authors say in the discussion (lines 521-524) that their data corroborates to literature that PTGS2 is regulated by NF-kB activation. However, the authors do not present any data regarding NF-kB activation, instead, they mention some unshown transcriptome data which would suggest NF-kB downregulation. Wouldn’t NF-kB downregulation afterward result in lower levels of PTGS2? That is why NF-kB activation and additional pathways must be better explored (writing/experimentally) by the authors before they propose such a sequence of events after infection with P or NP strains of leptospires, illustrated in figure9.

Reviewer #2: The review comments are finalized in uploaded documents.

Reviewer #3: In this work, the authors studied DC derived from human monocytes (MoDC) infected with pathogenic and saprophytic strains of leptospires, and their ability to activate CD4+ T cells in co-culture. They confirmed that compared to saprophytic strains, pathogenic Leptospira less activate MoDC and showed for the first time that pathogenic leptospires skew the DC/T-cell response to anti-inflammatory cytokines and Treg response. They also add some apoptosis, transcriptional and phagocytosis analyses.

Major comments 

This comparative study between pathogenic and saprophytic strains of leptospires is interesting since it provides the evidence that less MoDC activation by pathogenic Leptospira strains leads to a skewed anti-inflammatory CD4 T cell activation and T reg response. This is an important mechanism that could explain the deficient T cell-dependent adaptive response upon leptospirosis. 

However, this article suffers from different drawbacks and could be improved by:

 1) adding isotype controls. Indeed, FMO and isotypes were only included when appropriate (line 222) but results are not shown here. Yet, threshold of fluorescence should be done using isotypes and not unstained cells. This should be performed and, at least, shown in sup data. The analysis should be raw MFI instead of “relative MFI”. It would permit to visualize the basal expression of the different activation markers in non-infected cells. Also, why did the authors not study CD40 up-regulation?

2) confirming the MoDC apoptosis phenotype with additional experiments. Please add a positive control (staurosporine?) to be able to gate the apoptotic cells. Moreover, only a maximum of 10% of MoDCs infected with pathogenic strains undergo early apoptosis, and almost none are detected in late apoptosis: it would then be interesting to measure apoptosis with other methods (caspase 3 cleavage by WB, or Caspase 3/7 activity kit) to really assess apoptosis of MoDCs upon infection with pathogenic strains. A kinetic would be required too. Then the meaning of this phenotype regarding the MoDCs functions should be carefully discussed, avoiding overstatements and references not always appropriate (471-477). 

3) removing parts that are not essential for the purpose of this study (transcriptomic and phagocytosis). Indeed, the transcriptomic study is not properly explained nor presented. The interesting Cox2 result does not bring much to this study and could better be kept for another article, exploiting in detail the transcriptomic results. The “phagocytic” part is not properly conducted. Authors did not do gentamicin assays, so they probably only measured Leptospira-associated bacteria by cytometry. The “index” factor of phagocytosis is not appropriate to study live and motile bacteria. If intracellular Leptospira are observed by confocal microscopy (please add the non-infected MoDCs control), it does not mean phagocytosis, which relies on actin rearrangements that have not been evidenced by other groups who showed active entry of leptospires in macrophages, despite of cytochalasin treatment. 

4) editing/rewriting the manuscript and title to better highlight the novelty of this work. It would improve its readability, while avoiding overstatements (see minor comments). Moreover, in the manuscript (and especially in the discussion part), a lot of references are poorly corresponding to the text. The authors should carefully check each sentence and associated reference to avoid approximative citations. 

Minor comments

Title A more factual and informative title could be: “Pathogenic Leptospira skew MoDC-CD4 T cells response towards anti-inflammatory IL-10 and Treg instead of protective IFN-g and TH1 by saprophytic strains” or substituted by the Authors’ short title “Impaired functions of human-MoDCs and induction of Treg cell by pathogenic Leptospira”.

Abstract to rewrite to better show the novelty of the study. 

The first part of the study is a confirmation of published data obtained with other serovars, showing that pathogenic strains less activate MoDCs than saprophytes. The abstract should be reformulated to put the emphasis on the novel results showing that pathogenic leptospires in contrast to saprophytic strains, skew the DC/T response to anti-inflammatory cytokines and Treg response. 

The sentence about apoptosis seems overstated. 

Please reformulate the last sentence so that the reader can understand how this study may aid the vaccine development.

Introduction. Line 93: the sentence “Additionally, LPS of pathogenic leptospires reduced CD40 expression and cytokine production of DCs, partially via the TLR4 and TRIF pathways” is inaccurate: in this ref only bacteria and not LPS have been used, and the LPS effect on TLR4 and TRIF is only true in murine macrophages since in human cells leptospires do not stimulate TLR4. Therefore, the sentence should be reformulated to mention the human MoDCs, which are more activated upon stimulation with the saprophytic strains compared to the pathogenic strains.

Line 100: A similar study on MoDCs activation upon infection with saprophytic and pathogenic Leptospira strains has already been published (ref missing). Here the authors should focus and study the consequences on CD4 T cell response of pathogenic Leptospira limiting DC activation, compared to the effect of saprophytic that stimulate better the MoDC-T cell response. 

Lines 101-104: should be put in the abstract rather than at the end of introduction.

Result. The first part of the study is a confirmation of published data obtained with other serovars, showing that pathogenic strains less activate MoDCs compared to saprophytic. Please rephrase carefully: MoDCs exposed to pathogenic leptospires show less activation than those exposed to saprophytic leptospires. The cells are still activated by pathogenic strains, they are just less so, as has been shown in the literature.

Figure 1 and 2 could be presented in the same figure. 

Line 259: the title of paragraph is not accurate. It should be “Pathogenic leptospires less activate MO-DC maturation than saprophytic leptospires. 

Fig 1: why did you use “relative MFI”? You would beneficiate from showing MFI, and please add isotype controls or FMO. 

Line 299: no gentamicin elimination of leptospires for the phagocytic assay. Therefore, better speak about Leptospira associated bacteria and internalization.

Line 333: this is not “phagocytosis” observed in cytometry but Leptospira associated with cells

Line 385: the conclusion seems overstated. There is no demonstration of “inhibition”. Data showed that MoDCs stimulated with saprophytic strains induce T cell proliferation in contrast to MoDCs infected with pathogens. 

Line 403: after 24h of stimulation? 

Discussion. Please rewrite carefully the discussion., which is not very clear and often confusing. Please avoid discussing data not shown (line 521). Please insist on the anti-inflammatory and Treg aspect. 

Line 404: should be written “pathogenic strains induced “impaired” MoDC responses” instead of “does not induce…”

Line 478 should be written “by lower” instead of “by low”

Line 481-492 The paragraph about the discrepancy is confusing. Virulence and pathogenicity should not be confused.

Line 499-512: please rewrite since it is not clear.

line 502: the authors give some explanations about the known function of PGS2 and PTS2 (Cox2). It should be done earlier, ideally in the result section.

line 159: misspelling: should be written “stained” instead of strained instead of 

Line 549: Replace Th1 by Th2

Lines 551-553: overstatement, TH1 response still needs to be investigated in leptospirosis

In all the manuscript, figures legends and sup data, please write MoDCs are “infected” with Leptospira and keep the word “co-cultured” for co-culture experiments of MoDCs with CD4-T cells. It will greatly improve the understanding. 

Fig 9 My suggestion would be show to only the panel D, to better recapitulate the work. Please better explain the Th1/Th2/ Treg phenotypes; also in text line 548-550 since it is very confusing.

PLOS authors have the option to publish the peer review history of their article (what does this mean?). If published, this will include your full peer review and any attached files.

Reviewer #1: No

Reviewer #2: Yes: Suman Kundu

Reviewer #3: No
---

## [Decision Letter · Decision Letter 1]

10 Oct 2023

Dear Dr. Patarakul,

Thank you very much for submitting your manuscript "Impaired Functions of Human Monocyte-Derived Dendritic Cells and Induction of Regulatory T cells by Pathogenic Leptospira" for consideration at PLOS Neglected Tropical Diseases. As with all papers reviewed by the journal, your manuscript was reviewed by members of the editorial board and by several independent reviewers. The reviewers appreciated the attention to an important topic. Based on the reviews, we are likely to accept this manuscript for publication, providing that you modify the manuscript according to the review recommendations. 

While all three reviewers acknowledged the improved quality and clarity of the revised manuscript, Reviewers #1 and #3 raised a few important points that should be addressed before publication. Particularly, address the possible inconsistencies in figures 2 and S5 pointed out by Reviewer #3, explaining your rationale for presenting the data as currently shown in the manuscript or including additional complementary experiments that properly and fully satisfy the reviewer's inquires.

Sincerely,

Andre Alex Grassmann, PhD

Guest Editor

Elsio Wunder Jr

Section Editor

While all three reviewers acknowledged the improved quality and clarity of the revised manuscript, Reviewers #1 and #3 raised a few important points that should be addressed before publication. Particularly, address the possible inconsistencies in figures 2 and S5 pointed out by Reviewer #3, explaining your rationale for presenting the data as currently shown in the manuscript or including additional complementary experiments that properly and fully satisfy the reviewer's inquires.

Reviewer's Responses to Questions

**Key Review Criteria Required for Acceptance?**

**Methods**

-Are the objectives of the study clearly articulated with a clear testable hypothesis stated?

-Is the study design appropriate to address the stated objectives?

-Is the population clearly described and appropriate for the hypothesis being tested?

-Is the sample size sufficient to ensure adequate power to address the hypothesis being tested?

-Were correct statistical analysis used to support conclusions?

-Are there concerns about ethical or regulatory requirements being met?

Reviewer #1: (No Response)

Reviewer #2: (No Response)

Reviewer #3: T

**Results**

-Does the analysis presented match the analysis plan?

-Are the results clearly and completely presented?

-Are the figures (Tables, Images) of sufficient quality for clarity?

Reviewer #1: (No Response)

Reviewer #2: (No Response)

Reviewer #3: (No Response)

**Conclusions**

-Are the conclusions supported by the data presented?

-Are the limitations of analysis clearly described?

-Do the authors discuss how these data can be helpful to advance our understanding of the topic under study?

-Is public health relevance addressed?

Reviewer #1: (No Response)

Reviewer #2: (No Response)

Reviewer #3: (No Response)

**Editorial and Data Presentation Modifications?**

Reviewer #1: (No Response)

Reviewer #2: (No Response)

Reviewer #3: (No Response)

**Summary and General Comments**

Reviewer #1: I am happy that the authors addressed my concerns regarding the apoptosis assay by testing different MOIs and time points. The results of apoptosis are now much stronger and indisputable. However, the results showing that at 48h most MoDcs are dead makes me wonder if the authors should discuss in more detail this proportion of apoptotic cells in CD4 activation. The authors should also consider reviewing data from the literature that show the role played by PGE2 induced in and released by apoptotic cells that ultimately dampen immune responses (Hangai., et al, 2016) and, corroborate the author's findings. Furthermore, the authors' schematic representation completely neglected the key fact that pathogenic strains induce apoptosis.

*Hangai S, Ao T, Kimura Y, Matsuki K, Kawamura T, Negishi H, Nishio J, Kodama T, Taniguchi T, Yanai H. PGE2 induced in and released by dying cells functions as an inhibitory DAMP. Proc Natl Acad Sci U S A. 2016 Apr 5;113(14):3844-9. doi: 10.1073/pnas.1602023113. Epub 2016 Mar 21. PMID: 27001836; PMCID: PMC4833254.

Reviewer #2: The authors have made significant changes ad revised the manuscript. They have addressed most of the questions raised by the reviewers. The study is very thorough but major weakness of the study that it was performed in in vitro condition. Overall the study is significant and well executed.

Reviewer #3: The authors carefully answered and addressed all comments which greatly improved this manuscript, that is now easy to read with clear messages. 

However, I still have 2 remarks 

- 1) There is a major inconsistency about apoptosis and caspase 3 cleavage. 

At 48 hours after infection, the authors show by cytometry (with 3 dots corresponding to the 3 independent experiments with 3 different donors, nicely grouped) 80 % of apoptotic DC cells upon infection with pathogenic Leptospira Pomona, but the Western-Blots showing caspase 3 cleavage do not fully support this result. The WB in Fig2C is presented as a “representative Western blot …using samples from three donors to determine cleavage of caspase-3 in MoDCs. Then in, fig S5, a completely different WB (still labeled “representative” that I interpret as representative of the 2 other donors) shows that Leptospira strains (pathogenic and nonpathogenic) do not induce more cleavage of caspase 3 compared to the non-stimulated control. 

Therefore, the caspase 3 dependent apoptosis experiment is not convincing. The authors should not crop the image of WB to show the upper non-cleaved band of pro-caspase 3, which is important to assess the efficiency of the positive control. The GAPDH controls can be put below. The authors could have also provided the MTS assay at 48h in Fig S4. I think the data about apoptosis should be more robust (please add at least 5 more donors for WB and cytometry, before concluding of a strategy of leptospires to induce apoptosis to avoid the T-cell response. 

-2) The change of statistical representation. I know that the statistics with R and letters seems at first difficult to understand, but it is indeed better to represent complete statistics and avoid fastidious representation with overlapping lines.

PLOS authors have the option to publish the peer review history of their article (what does this mean?). If published, this will include your full peer review and any attached files.

Reviewer #1: No

Reviewer #2: Yes: Suman Kundu

Reviewer #3: No

Figure Files:

Data Requirements:

Reproducibility:

References

---

## [Editor Report · Decision Letter 2]

2 Nov 2023

Dear Dr. Patarakul,

Thank you very much for submitting your manuscript "Impaired Functions of Human Monocyte-Derived Dendritic Cells and Induction of Regulatory T cells by Pathogenic Leptospira" for consideration at PLOS Neglected Tropical Diseases. As with all papers reviewed by the journal, your manuscript was reviewed by members of the editorial board and by several independent reviewers. The reviewers appreciated the attention to an important topic. Based on the reviews, we are likely to accept this manuscript for publication, providing that you modify the manuscript according to the review recommendations. 

After thorough revision by the Guest Editor and considering the previous feedback from Reviewer #3 as well as the Authors' response, the Editor is requesting modifications to the manuscript. These textual modifications should acknowledge all the limitations in the apoptosis experiments presented in the current study, with special emphasis on the small sample size used for these experiments. As for the statistical representation of significance, the Guest Editor prefers the "letter-based system," which was the Authors' initial choice and also seems to be preferred by Reviewer #3.

Sincerely,

Andre Alex Grassmann, PhD

Guest Editor

Elsio Wunder Jr

Section Editor

After thorough revision by the Guest Editor and considering the previous feedback from Reviewer #3 as well as the Authors' response, the Editor is requesting modifications to the manuscript. These textual modifications should acknowledge all the limitations in the apoptosis experiments presented in the current study, with special emphasis on the small sample size used for these experiments. As for the statistical representation of significance, the Guest Editor prefers the "letter-based system," which was the Authors' initial choice and also seems to be preferred by Reviewer #3.

Figure Files:

Data Requirements:

Reproducibility:

References

---

## [Editor Report · Decision Letter 3]

8 Nov 2023

Dear Dr. Patarakul,

We are pleased to inform you that your manuscript 'Impaired Functions of Human Monocyte-Derived Dendritic Cells and Induction of Regulatory T cells by Pathogenic Leptospira' has been provisionally accepted for publication in PLOS Neglected Tropical Diseases.

Best regards,

Andre Alex Grassmann, PhD

Guest Editor

Elsio Wunder Jr

Section Editor

---

## [Editor Report · Acceptance letter]

15 Nov 2023

Dear Dr. Patarakul,

We are delighted to inform you that your manuscript, "Impaired Functions of Human Monocyte-Derived Dendritic Cells and Induction of Regulatory T cells by Pathogenic Leptospira," has been formally accepted for publication in PLOS Neglected Tropical Diseases.

Best regards,

Shaden Kamhawi

co-Editor-in-Chief

Paul Brindley

co-Editor-in-Chief
